

# Straw incorporation increases crop yield and soil organic carbon sequestration but varies under different natural conditions and farming practices in China: a system analysis

Xiao Han[1‡], Cong Xu[1‡], Jennifer A. J. Dungait[2], Roland Bol[3], Xiaojie Wang[1], Wenliang Wu[1], and Fanqiao Meng[1]

[1]Beijing Key Laboratory of Biodiversity and Organic Farming, College of Resources and Environmental Sciences, China Agricultural University, Beijing, 100193, China
[2]Sustainable Agriculture Sciences, Rothamsted Research, North Wyke, Okehampton, Devon EX20 2SB, UK
[3]Institute of Bio- and Geosciences, Agrosphere (IBG-3), Forschungszentrum Jülich GmbH, 52425 Jülich, Germany

[‡] These authors contributed equally to this work

*Correspondence to:* Fanqiao Meng (mengfq@cau.edu.cn)

**Abstract.** Loss of soil organic carbon (SOC) from agricultural soils is a key indicator of soil degradation associated with reductions in net primary productivity in crop production systems worldwide. Simple technical and locally appropriate solutions are required for farmers to increase SOC and to improve cropland management. In the last 30 years, straw incorporation has gradually been implemented across China in the context of agricultural intensification and rural livelihood improvement. A meta-analysis of data published before the end of 2016 was undertaken to investigate the effects of straw incorporation on crop production and SOC sequestration. The results of 68 experimental studies throughout China in different edaphic, climate regions and under different farming regimes were analyzed. Compared with straw removal, straw incorporation significantly sequestered SOC (0−20 cm depth) at the rate of 0.35 (range 0.31−0.40) Mg C ha$^{-1}$ yr$^{-1}$, increased crop grain yield by 13.4% (range 9.3%–18.4%) and had a conversion efficiency of the applied straw-C as 16% ± 2% across the whole of China. The combined straw incorporation at the rate of 3 Mg C ha$^{-1}$ yr$^{-1}$ with mineral fertilizer of 200–400 kg N ha$^{-1}$ yr$^{-1}$ was demonstrated to be the best combination for farmers to use with crop yield increased by 32.7% (range 17.9%–56.4%) and SOC sequestrated by the rate of 0.85 (range 0.54–1.15) Mg C ha$^{-1}$ yr$^{-1}$. Straw incorporation achieved higher SOC sequestration rate and crop yield increment when applied to clay soils, under high cropping intensities, and in areas like Northeast China where the soil is being degraded. SOC responses were the greatest in the initial starting phase of straw incorporation and then declined and finally were negligible after 28−62 years, however, crop yield responses were initially low and then increased reaching their highest level at 11–15 years after straw incorporation. Overall, our study confirmed that straw incorporation did create a positive feedback loop of SOC enhancement together with increased crop production, and this is of great practical significance to straw management as agricultural intensifies in China and other regions in the world with different climate conditions.



## 1 Introduction

Around a quarter of China's land territory (or more than 2 million km$^2$) is affected by soil degradation associated with the loss of around net primary productivity equating to nearly 60 billion Mg carbon (C) over 23 years (Bai et al., 2008). The considerable impact of soil degradation on crop production in China

and worldwide points to the need for solutions appropriate to location-specific agro-ecological conditions and farming systems (Bindraban et al., 2012). Soil organic carbon (SOC) loss is a key indicator of soil degradation that is accelerated by land use (Erb et al., 2016; Liu et al., 2018), and is widely associated with cultivation (Dungait et al., 2012; Amundson et al., 2015). Thus, management to enhance SOC to potentially rejuvenate degraded agricultural soils thereby improving soil fertility and increasing crop

yield (Smith et al., 2012), whilst sequestering soil carbon to mitigate climate change (Meinshausen et al., 2009), is a win: win scenario that maintains the integrity of agricultural ecosystems (Power, 2010).

Like many degraded arable soils across the world, cropland soils in China commonly have poor SOC concentrations (12.0−12.7 g kg$^{-1}$, 0−20 cm; Yan et al., 2011) which suggests a substantial potential for C sequestration (25–37 billion Mg C yr$^{-1}$) if management is changed to rebuild SOC stocks in cultivated

soil (Lal, 2002). Since the start of the reform policies in 1978, China has experienced a series of rapid agricultural intensification processes, which were characterized by a main farm management practices mostly involving high mineral fertilization rate (e.g. > 400 kg N ha$^{-1}$ yr$^{-1}$; Ju et al., 2004), frequent irrigation events (Kong et al., 2016) and intensification of mechanization (Zhang et al., 2017). This greatly increased not only the grain yield, but also straw yields to a > 0.6 billion Mg straw yr$^{-1}$ from three

crops of maize, wheat, and rice (Shi et al., 2014). Crop straw was widely harvested for fuel but, with the improvement in rural livelihoods after the 1990s, farmers have tended to switch to electricity, liquid gas or coal (Zhang et al., 2017), introducing challenges for managing large amounts of 'waste' straw (Kong et al., 2014). The recently renewed recognition of the importance of SOC for soil health and quality has encouraged straw incorporation as a simple and environmentally-friendly measure to effectively enhance

cropland SOC levels (Pan et al., 2010) and to improve crop production (Zhao *et al*., 2015).

Differences in climatic and edaphic conditions (Bolinder et al., 2007), fertilization strategies (Khan et al., 2007), cropping regimes (Huang et al., 2012) and duration of straw incorporation (Lehtinen et al., 2014) have resulted in large spatial and temporal variations in the effects of straw incorporation on SOC and crop yield in China (Yu et al., 2012; Li et al., 2003). Extensive large field experiments have been

conducted since the 1980s to study the effect of straw incorporation (e.g. Zhang et al., 2014; Gong et al., 2009; Cai and Qin, 2006), and these have helped to achieve a more systematic understanding of the benefits of straw incorporation. Integration of the results of studies covering different regions and under varied farming practices also assist an effective examination of the underlying mechanism of straw incorporation on SOC, (e.g. SOC conversion efficiency; Kong et al., 2005; Wang et al., 2015) and crop

yield. This novel information could provide the scientific basic support for the development of sound policy for straw management at regional and governmental levels (Ministry of Agriculture−PRC, 2013, 2015).

We selected meta-analysis to test the hypothesis that straw incorporation increases SOC stocks and crop yields in China, since it is an effective proven statistical method to quantitatively integrate the results of

numerous individual studies and from that to draw general conclusions at a larger scale (Gurevitch et al.,



2001; Chivenge et al., 2011). To date, several meta-analyses have reported on the effects of straw incorporation on SOC/crop yield in China's arable soils (e.g. Lu et al., 2009; Tian et al., 2015; Wang et al., 2015; Zhao et al., 2015). For instance, Lu et al. (2009) reported that straw incorporation could sequester 9.76 billion Mg C yr$^{-1}$ in China's cropland; Zhao et al. (2015) reported that straw incorporation improved crop yield by 7% across China. However, few of these studies presented the effects of straw incorporation in different climatic and edaphic regions (Lu et al., 2009; Zhao et al., 2015); or addressed simultaneously the responses of crop yield and SOC to straw incorporation (Wang et al., 2015), which commonly are interactively influenced by many environmental factors and also farming management measures (Pan et al., 2009; Loveland and Webb, 2003). The poor reporting of straw C conversion efficiencies (Tian et al., 2015; Lu et al., 2009) also weakens the practicability of some of the management related conclusions in policy development. To overcome the limitation of previous meta-analysis studies, we conducted a new meta-analysis of field experiments carried out over the last 30 years in China. We aimed to: (i) quantify the responses of SOC and crop yield to straw incorporation at regional and national scales; (ii) calculate the conversion efficiency of straw-C to SOC; and (iii) assess the effects of major factors, i.e. soil properties (texture, initial SOC content), climate conditions (temperature, rainfall) and farming practices (straw quantity and type, incorporation duration, N fertilizer and cropping system) on the efficacy of straw incorporation.

## 2 Materials and methods

### 2.1 Data source

A survey of peer-reviewed research papers published in China before 31 December 2016 was conducted in using two bibliographic databases: Web of Knowledge and Chinese Journal Databases (CNKI). The keywords "soil organic carbon", "straw incorporation" and "straw return" were used. To be included in the meta-analysis, a study had to meet the following criteria: (i) it was based on a field experiment lasting for more than 3 years, with a known starting year; (ii) experimental treatments were replicated; (iii) experiments had paired treatments of both straw incorporation and straw removal; and, (iv) cropping systems included at least one crop of rice, maize or wheat. A total of 68 papers (Table S1), consisting of 70 long-term field experiment sites (Fig. 1) and 172 paired SOC data, did meet the criteria for inclusion in our experiment. Of the 68 papers, 33 also presented crop yields.

Information on soil properties (texture, initial SOC content, bulk density), climate (temperature, precipitation), farming practices (land use, N fertilization, crop type, crop frequency, C and nutrient contents of straw, duration of straw incorporation) was also collected. The SOC content or stock and crop yields were obtained directly from tables and/or text of the papers or extracted from the figures using graph digitizing software (GetData Graph Digitizer V2.25; http://getdata-graph-digitizer.com/). For studies where SOC content were reported without bulk density, SOC stock was calculated using Eq. (1):

$$SOC\ stock\ (\text{Mg C ha}^{-1}) = SOC \times BD \times H \times 0.1 \tag{1}$$

where $SOC$ is SOC content (g kg$^{-1}$), $BD$ is the soil bulk density (BD) (g cm$^{-3}$), $H$ is the thickness of the soil layer (0−20 cm) and 0.1 is a constant to adjust the units. The SOC stocks were computed to 20 cm





depth. In those studies that only reported soil organic matter content, we estimated SOC content as 58% of the soil organic matter. For the studies in which BD was not available, we estimated the BD for paddy or paddy–upland soil using Eq. (2) (Pan et al., 2003):

$$BD = -0.22 \times \ln(SOC) + 1.78 \tag{2}$$

and for upland soil, the BD was estimated using Eq. (3) (Song et al., 2005):

$$BD = 1.377 \times \text{Exp}(-0.0048 \times SOC) \tag{3}$$

For studies that did not report the quantity of incorporated straw C, this was calculated by multiplying the straw C content (39.9% for wheat, 44.4% for maize and 41.8% for rice) with the amount of straw incorporated (NATEC, 1999). The amount of N, P, and K was similarly computed for the incorporated

straw.

To distinguish between the sources of variation for the responses of SOC and crop yield to straw incorporation, the paired measurements were further subdivided into subgroups according to the categorical variables listed in Table 1. Annual fertilizer N input in the studies ranged from 0 to 720 kg N ha$^{-1}$ yr$^{-1}$ and was separated into three levels. The $> 400$ kg N ha$^{-1}$ yr$^{-1}$ and $200-400$ kg N ha$^{-1}$ yr$^{-1}$ ranges

represent the current farmer's fertilizer N practices and the optimized fertilizer N rates, respectively, whereas $< 200$ kg N ha$^{-1}$ yr$^{-1}$ a low N fertilization level (Zhang et al., 2017; Ju et al., 2004). Mean annual precipitation (MAP) and mean annual temperature (MAT) ranged from 117 to 1788 mm and from 0.9 to 18.4°C, respectively. The classifications of MAP and MAT in the meta-analysis were based on FAO guidelines for agro-climatic zoning (Fischer et al., 2002). Mainland China was divided into four regions

according to the geographic location, climate conditions, and farming practices: Northeast China (NEC), North China (NC), Northwest China (NWC) and South China (SC). Detailed information for each region is listed in Table 2. Other categorized variables were crop frequency (number of crops per year, i.e. single, double and triple crops), land use type (i.e. paddy, upland and paddy-upland soils) and straw type (i.e. rice, wheat and maize straws).

**2.2 Data analysis**

**2.2.1 Responses of crop yield to straw incorporation**

Effect size is an index that reflects the magnitude of treatment (crop straw) effect in comparison with a reference treatment (Borenstein et al., 2009). The effect size of each observation (comparison between straw incorporation and straw removal, in our study) for crop yield was calculated as the natural log of

the response ratio (ln$R$) (Rosenberg et al., 2000), as in Eq. (4):

$$\ln R = \ln \frac{X_e}{X_c} \tag{4}$$

where $X_e$ is the mean crop grain yield of the straw incorporation treatment and $X_c$ is the mean grain yield of the control (straw removal). The relative change in crop yields following straw incorporation was also calculated as $(R-1) \times 100\%$ (Chivenge et al., 2011). Positive values of relative change indicated a

promotion effect of straw incorporation on crop production and *vice versa*.





### 2.2.2 Responses of SOC to straw incorporation

The effect size of SOC was expressed as an annual SOC sequestration rate (Mg C ha$^{-1}$ yr$^{-1}$), which was calculated by Eq. (5):

$$Annual\ SOC\ sequestration\ rate\ (\text{Mg C ha}^{-1}\ \text{yr}^{-1}) = \frac{(Dsoct - Dsoci) - (Dsoct' - Dsoci')}{duration} \qquad (5)$$

where $Dsoct$ and $Dsoct'$ are SOC stock for the final year of experimental straw incorporation and straw removal treatments, respectively; and $Dsoci$ and $Dsoci'$ are SOC stock for the initial year of straw incorporation and straw removal treatments, respectively. A positive value of annual SOC sequestration rate indicates the SOC stock increase due to straw incorporation and a negative difference indicates the opposite effect.

**2.3 Meta-analysis**

A meta-analysis of the random effect model was performed and analyzed using MetaWin 2.1 software (Rosenberg et al., 2000). As standard deviations were rarely available in the selected literature, but still be able to include as many studies as possible, an unweighted analysis was adopted (Hedges et al., 1999; Rosenberg et al., 2000). We used bootstrapping (4999 iterations) to generate the mean effect size and

bias-corrected 95% confidence intervals (95% CIs) for each categorical variable. Mean effect sizes were considered to be significantly different if the 95% CIs did not overlap with each other, and were considered to be significantly different from control if their 95% CIs did not overlap with zero (Chivenge et al., 2011). We accepted that the mean effect sizes of the categories to be significantly different between the levels of the factors if the $P$ values of the between-group heterogeneity ($Q_b$) were less than the 0.05

level ($P < 0.05$).

**2.4 Regression analysis**

A stepwise regression analysis was applied to analyze the relationship between SOC contents, the input rate of total nutrients (N, P$_2$O$_5$, K$_2$O) and crop yields. Regression analysis was also used to examine the SOC responses to experimental factors (i.e. straw C input rate, experiment duration and initial SOC

content). The relationship between yield response to straw incorporation and control yield was also examined. All regression analyses were performed using SPSS version 20.0 (SPSS Inc., Chicago, USA), and the results were considered statistically significant if $P < 0.05$.

**3 Results**

**3.1 SOC and crop yield**

A significant positive linear regression was determined between SOC content and crop yield (Fig. 2; $P < 0.05$). Stepwise regression analysis also revealed a significant linear relationship between crop yield and SOC, in which the factor of fertilization was considered, $Yield$ (Mg ha$^{-1}$ yr$^{-1}$) = 0.933 + 0.267 × $SOC$ (g kg$^{-1}$) + 0.008 × $N\ fertilizer$ (kg ha$^{-1}$ yr$^{-1}$) + 0.010 × $K\ fertilizer$ (kg ha$^{-1}$ yr$^{-1}$) (R$^2$ = 0.69, $P < 0.01$, n





= 100). This indicated SOC content could explain 42% of yield variations while SOC content and fertilizer input altogether explained 69%. Overall, an increase of 1 g kg$^{-1}$ SOC content could improve crop yield by 267−414 kg ha$^{-1}$ yr$^{-1}$, if converted to SOC stock (20 cm depth), the crop yield increment would be 101−157 kg ha$^{-1}$ yr$^{-1}$ (with soil BD assumed to be 1.32 g cm$^{-3}$ Han et al., 2012).

**3.2 Responses of crop yield to straw incorporation**

Straw incorporation significantly increased annual crop yield by 13.4% (range 9.3%–18.4%, 95% CI) relative to straw removal (Fig. 3a). The yield responses to straw incorporation were however different in the four different regions of China (Fig. 3b). The greatest yield increase corresponding to straw incorporation was observed in NEC (mean 26.8%, range 18.1%–38.2%), compared with SC (mean 11.6%, range 7.3%–17.7%) and NC (mean 9.8%, range 3%–26.7%), and the poorest response in NWC (mean 7.3%, range 1.8%–13.6%).

Yield increase was positively related to the duration of straw incorporation for the first 15 years, and increased from 4.9% (range 3.0%–7.5%) after 3−5 years, to 12.3% (5.1%–20.7%) after 6−10 years, and to 18.6% (range 12.4%–26.5%) after 11−15 years. After 15-year, the yield increase (12.6%, 5.1%–20.4%) tended to decline to a level similar to that reported for 6−10 years (Fig. 3d).

Crop frequency significantly affected the relationship between straw incorporation and crop yield: the increment of grain yield was greater in single (mean 15.1%, range 9.9%–21.2%) and double cropping systems (mean 12.5%, range 7.1%–20.7%), compared with triple cropping systems (mean 4.5%, range 2.3%–8.3%) (Fig. 3c). Meanwhile, yield increases greatly varied between crops: 8.7% (range 4.1%–13.5%), 20.8% (range 12.8%–31.0%) and 10.4% (range 6.6%–15.3%) for wheat, maize and rice, respectively (Fig. 4). Yield response was greater when the control (straw removal) yield was low and, as the yield of control increased, the yield response to straw incorporation became smaller (Fig. 4).

There was a significant ($P < 0.05$) positive relationship between the amount of straw incorporated and crop yield, with high levels of straw input corresponding to mean increases of 28.4% (range 18.6%–40.9%) compared to low (mean 15.0%, range 9.1%–22.5%) and medium (mean 6.9%, range 2.3%–14.5%) straw input (Table 3). Crop yield responses generally increase in response to the combination of straw application with mineral N application, i.e. increasing from 11.5% (range 6.2%–18.0%) to 18.4% (range 11.9%–27.6%) when the N-application rate increased from 0−200 to 200−400 kg N ha$^{-1}$ yr$^{-1}$. However, the highest level of N fertilizer (> 400 kg N ha$^{-1}$ yr$^{-1}$) did not result in a significant additional yield increase (mean 18.8%, range, 1.6%–54.2%) as compared to 200−400 kg ha$^{-1}$ yr$^{-1}$ (mean 18.4%, range 11.9%–27.6%) (Table 3).

**3.3 Responses of SOC to straw incorporation**

Annual SOC sequestration in response to straw incorporation was enhanced, with an average rate of 0.35 (range 0.31−0.40) Mg C ha$^{-1}$ yr$^{-1}$ (Fig. 5a). A significant effect of between-group heterogeneity ($Q_b$) was found between the four geographical regions (Fig. 5b; $P < 0.05$), and between different straw incorporation duration, crop frequency and soil texture (Fig. 5c, e, i; $P < 0.05$), but not for land use type, straw type, MAT or MAP (Fig. 5d, f, g, h; $P > 0.05$). Compared to the control treatments (straw removal),





the greatest SOC sequestration rate were recorded in NEC (mean 0.57, range 0.41−0.77 Mg C ha$^{-1}$ yr$^{-1}$), followed by SC (mean 0.36, range 0.30−0.43 Mg C ha$^{-1}$ yr$^{-1}$), NC (mean 0.33, range 0.25−0.40 Mg C ha$^{-1}$ yr$^{-1}$) and NWC (mean 0.19, range 0.14−0.25 Mg C ha$^{-1}$ yr$^{-1}$) (Fig. 5b). The annual SOC sequestration rates were significant greater in the shortest time interval (3−10 years) after straw

incorporation began (mean 0.53, range 0.44−0.63 Mg C ha$^{-1}$ yr$^{-1}$), compared to the medium-term (10−20 years; mean 0.29, range 0.23−0.37 Mg C ha$^{-1}$ yr$^{-1}$) or long-term (> 20 years; mean 0.17, range 0.13−0.21 Mg C ha$^{-1}$ yr$^{-1}$) (Fig. 5c).

The effect of straw incorporation on SOC sequestration varied between different crop frequencies in the order: triple (mean 0.51, range 0.37−0.67 Mg C ha$^{-1}$ yr$^{-1}$) > single (mean 0.41, range 0.31−0.53 Mg C

ha$^{-1}$ yr$^{-1}$) > double (mean 0.28, range 0.24−0.32 Mg C ha$^{-1}$ yr$^{-1}$). The SOC sequestration after straw incorporation in clay soils (mean 0.43, range 0.36−0.52 Mg C ha$^{-1}$ yr$^{-1}$) was significantly ($P < 0.05$) increased compared with loam soil (mean 0.25, range 0.20−0.30 Mg C ha$^{-1}$ yr$^{-1}$), but was not different to silt loam (mean 0.37, range 0.24−0.53 Mg C ha$^{-1}$ yr$^{-1}$) or sandy loam (mean 0.32, range 0.25−0.39, Mg C ha$^{-1}$ yr$^{-1}$) soils (Fig. 5i; $P > 0.05$). Rice straw and maize straw tended to sequester more SOC than

wheat straw, but the difference was not statistically significant (Fig. 5f; $P > 0.05$).

The mean overall SOC sequestration rates were 0.20 (range 0.16−0.25) Mg C ha$^{-1}$ yr$^{-1}$ under the lowest straw C input level (< 1.5 Mg C ha$^{-1}$ yr$^{-1}$) but increased significantly ($P < 0.05$) to 0.70 (range 0.53−0.88) Mg C ha$^{-1}$ yr$^{-1}$ under the highest straw C input (> 3 Mg C ha$^{-1}$ yr$^{-1}$) (Table 3). Nitrogen fertilizer input rate significantly positively ($P < 0.01$) increased SOC responses to straw incorporation, i.e. the average

annual SOC sequestration rate increased from 0.27 (range 0.22−0.32) to 0.69 (range 0.53−0.81) Mg C ha$^{-1}$ yr$^{-1}$ when the N application rates increased from 0−200 to > 400 kg N ha$^{-1}$ yr$^{-1}$. Interestingly, we found significant ($P < 0.05$) positive interaction of straw incorporation with fertilizer N input on SOC accrual (Table 3).

### 3.4 Relationships between SOC sequestered and straw input, experiment duration, and initial SOC
content

The meta-analysis revealed a significant positive linear relationship between annual SOC sequestration rate and straw C input across China (Fig. 6; $P < 0.05$). Based on the straw C conversion efficiency derived from the regression equations (slope of the linear correlation equation (Kong et al., 2005), the conversion efficiency of straw C to SOC was 16% ± 2% (mean ± standard error) for the whole of China, 30% ± 4%

in NEC, 11% ± 3% in NC, 8% ± 2% in NWC and 13% ± 4% in SC (Fig. 6a, b, c, d, e).

There was a significant logarithmic relationship between annual SOC sequestration rate and straw incorporation duration (Fig. 7; $P < 0.05$). A quick decline in SOC sequestration rate was observed after the initial stage of straw incorporation, especially in NEC and SC, and then the SOC sequestration rate decreased to a steady state. The SOC increment diminished to negligible after 46, 26, 35, 63 and 55 years

of straw incorporation in the whole nation and NEC, NC, NWC, and SC respectively (Fig. 7a, b, c, d, e).

### 4 Discussion

The results of the meta-analysis suggest that straw incorporation increase SOC stocks and crop yields in





experimental trials across China, regardless of the climate or soil type. Short term gains were significant, but the largest response was observed up to 15 years after straw incorporation began. This conclusion is based on a wide range of soils and climate conditions and suggests that farmers across the world may be able to use this simple management tool to increase their outputs by improving the quality of their soil, whilst mitigating climate change.

### 4.1 Increase of SOC by straw incorporation

The straw incorporation in the experimental systems reviewed across China significantly enhanced SOC at an average rate of 0.35 Mg C ha$^{-1}$ yr$^{-1}$ (relative to straw removal; Fig. 5a) regardless of the straw type (i.e. source crop) ($P > 0.05$), also observed by (Wang et al., 2016). This estimate is comparable to SOC sequestration rates reported after reviews for the croplands of the USA (0.1–0.3 Mg C ha$^{-1}$ yr$^{-1}$; IPCC, 2000), but only half that estimation for the EU (0.7 Mg C ha$^{-1}$ yr$^{-1}$; Smith, 2004). Like Tian et al. (2015), through meta-analysis we observed that SOC contents increased regardless of the initial SOC contents (data not shown), and that a rapid increase in SOC density occurred in the first two decades rather than in later periods (26−63 years) of straw incorporation (Fig. 5c, 7), which suggest that an equilibrium between C input and decomposition had been reached but only after decades. This supports the need for continued investment in long-term field experimentation to provide robust information about the impact of management in agroecosystems to inform farmer decision-making and policy (Macdonald et al., 2015). Straw incorporation provides a C direct source for the formation of SOC (Blanco-Canqui and Lal, 2009; Mulumba and Lal, 2008). and the greater the annual straw-C input rate, the faster SOC sequestration increased (Table 3, Fig. 6), as previously described (Kong et al., 2005; Maillard and Angers, 2014; Liu et al., 2014). Sequestration rates were increased where greater amounts of annual above-ground crop residues were input under double and triple cropping regimes (Fig. 5e; $P < 0.05$), also observed by (West and Post, 2002; Blanco-Canqui and Lal, 2009).

Nitrogen fertilization enhanced the effect of straw incorporation (Table 3), presumably because straw has a high C: N ratio and much of the N added at lower rates (up to 400 kg ha$^{-1}$) when straw input was high (> 3 Mg C ha$^{-1}$) was immobilized, at least in the short term (Singh et al., 1999). Furthermore, N fertilizer addition can enhance both above and belowground biomass production (Ladha et al., 2011; Neff et al., 2002; Kuzyakov and Domanski, 2000), increasing the input of crop roots to stable SOC pools (Gong et al., 2012). Addition of organic matter is generally correlated with improved soil structure, e.g. aggregate formation and stability (Blanco-Canqui and Lal, 2009; Mulumba and Lal, 2008), which physically protects SOC from decomposition (Six et al., 2002). Clay soils have a propensity to stabilize SOC by providing chemical and physical protection (Six et al., 2002), as borne out by this study wherein sequestration rates in clay were greater than in loam soils (Fig. 5i).

A significant spatial variation in straw incorporation effect on SOC was observed between the four regions of China defined for analysis (Fig. 5b; $P < 0.05$). Straw conversion efficiency (30%; Fig. 5b;) and annual SOC sequestration rate (0.57 Mg C ha$^{-1}$ yr$^{-1}$; Fig. 6) were notably greater in NEC compared with the other three regions. This is likely attributable to the colder climate (mean annual temperature of 0.9−8.1°C; Table S1) which restricts SOC decomposition (Karhu et al., 2014). Moreover, SOC stocks in NEC are rapidly declining (from 48.7 in the 1980s to 42.4 Mg C ha$^{-1}$ in 2006; Pan et al., 2010), because



of large-scale land reclamation of wetland to cropland since the 1970s (Gao et al., 2015), compared to large-scale increases in SOC in the majority of croplands in NC, NWC and SC (Yu et al., 2012). According to a farmer survey across China carried out by (Zhang et al., 2017), the percentage of straw residue retention was only 8.7% in NEC, while 32.7% burnt in the field. Our results highlight the need

to encourage the local farmers to incorporate straw to maintain SOC stocks in NEC.

The impact of land use, MAT, and MAP on straw-induced SOC sequestration was not statistically significant (Fig. 5; $P > 0.05$), in agreement with previous meta-analyses Liu et al. (2014) and Huang et al. (2012). Since alternative wetting and drying has been widely applied as a common practice to improve crop yield in paddy soils in China (Zhao et al., 2013), this wetting and drying cycles stimulate microbial

activity and increases organic matter mineralization during the mid-season drainage period (Mikha et al., 2005) and leads to a less stable form of SOC in paddy soils (Cui et al., 2012). As arable cropping systems are complex ecosystems controlled by both natural factors and farming practices (Lohila et al., 2003; Song et al., 2005), the direct effect of MAT and MAP might be largely overwhelmed and overridden by farming practices (Maillard and Angers, 2014; Pei et al., 2016). For instance, the MAP only ranges from

455 to 821 mm in NC (Table S1), much less than the variations of irrigation (0 to 667 mm yr$^{-1}$; Liao et al., 2015). Our results imply that improving SOC through straw incorporation might be also applicable to other regions in the world with different climate conditions or land use.

### 4.2 Straw incorporation increased crop yield

Straw incorporation significantly increased the overall crop yield by 13.4% compared to straw removal.

This yield increase is similar in magnitude to a recent global analysis (12%; Liu *et al.* 2014), but larger than previous meta-analyses of published data from China (up to 9% increase; Zhao et al., 2015; Wang et al., 2015; Huang et al., 2013; Xu et al., 2017) and those of the EU (6% increase; Lehtinen et al., 2014). The lower estimates reported in previous studies focused on shorter time periods, e.g. 8−12 years in Zhao et al. (2015), with this new analysis showing the greatest benefits of straw incorporation for crop yield

after 11−15 years (Fig. 3d); or more limited geographical spread, e.g. only NC was considered in (Xu et al., 2017) and only three sites from NEC were included in the analysis by Wang et al. (2015) and Huang et al. (2013). According to the current cereal production level in China (616 million Mg; NBSC, 2016), this straw-induced yield increment (13.4%) results of additional 82.6 million Mg of agricultural products. At the current per-capita food consumption of 388 kg yr$^{-1}$ (Central People's Government−PRC, 2008),

this could feed 0.2 billion people or 15% of China's population.

As discussed above, straw incorporation significantly increases SOC (Wang et al., 2015; Liu et al., 2014), which is a key determinant in crop production (Singh et al., 2002). In the current meta-analysis, regression analysis revealed that each increase of 1 Mg C ha$^{-1}$ SOC in the root zone could improve crop yield by 101−157 kg ha$^{-1}$ yr$^{-1}$ (Fig. 2), which fell within the range of 30−300 kg ha$^{-1}$ yr$^{-1}$ obtained for

Asia region by Lal (2013). Straw incorporation does also reduce soil compaction (Soane, 1990), moderates soil temperature (Li et al., 2013) and retains soil water in the plow layer (Zhang et al., 2014), and immobilizes N for later release in the growing season (Hansen et al., 2015), all of which may contribute further to promote crop production. Straw contains various macro- and micro-nutrients, incorporated from the soil and foliar applications of fertilizer during plant growth, which can contribute



to the nutrient budget of farms if returned to the soil (Lal, 2013). In the current study, the average annual N, P and K nutrients derived from straw residues were 35 kg N ha$^{-1}$ yr$^{-1}$, 13 kg P$_2$O$_5$ ha$^{-1}$ yr$^{-1}$ and 78 kg K$_2$O ha$^{-1}$ yr$^{-1}$, which accounted for 15%, 11% and 52% of the average annual mineral N, P and K input, respectively. Wang et al. (2015) reported that the estimated annual straw N input rate (39.6 kg N ha$^{-1}$

yr$^{-1}$) significantly and positively contributed to the crop yield increase. Similarly, Singh et al. (2002) suggested that crop residue recycling determines the soil K balance and thus affects crop production substantially. Our analysis did observe a strong indication that maize yields benefited more (20.8%) from straw incorporation than those of wheat (8.7%) (Fig. 4). This observation agrees with the study of Hijbeek et al. (2017), who found that maize benefits significantly from organic inputs than those of wheat

or barley. The most likely reason for this difference was that, compared to wheat, maize is mostly grown require with higher temperature and precipitation (Tan et al., 2017), and these conditions favor a fasten straw decomposition and also result in a more rapid and abundant nutrient release (Hartmann et al., 2014; Ladha et al., 2011).

Our study found that only when straw incorporation duration was < 15 years, yield responses tended to

increase with time (Fig. 3d). This response might be related to that for the greater soil fertility and yield sustainability to be achieved, a long-term adoption of straw incorporation in arable cropping management is required (Bi et al., 2009). Indeed, the yield increase was relatively low in the initial stage (5% in the first 3−5 years; Fig. 3d) but increased thereafter. However, unlike several other studies (Wang et al., 2015; Zhao et al., 2015), we observed that the yield increase tended to decline after a 15-year timespan of the

adoption of straw incorporation (Fig. 3d). The crop yield response was lower under the triple cropping system compared with that under the single or double cropping systems (Fig. 3c). In China, the areas where triple cropping was adopted usually received adequate rainfall (MAP > 1000 mm, Table S1) supporting good rates of crop production (i.e. 13.5 Mg ha$^{-1}$ yr$^{-1}$ versus 9.2 and 4.9 Mg ha$^{-1}$ yr$^{-1}$ in double and single cropping, respectively, in our database); thus straw-induced soil water retention might

contribute little benefit for crop production in the region (Raffa et al., 2015) compared with the drier regions.

The probable N immobilization effect of straw incorporation discussed above has potential benefits for the crop and environment (e.g. reduced losses by leaching and N$_2$O emissions; Meng et al., 2016) through improved nitrogen use efficiency (Yao et al., 2017). In the current meta-analysis, under the N fertilizer

level of 200−400 and > 400 kg N ha$^{-1}$ yr$^{-1}$, which was widely adopted in arable land in China (Ju et al., 2004), 18% of yield increase due to straw incorporation was observed (Table 3). This emphasized that straw incorporation is an effective measure for both improving crop yield, as well as having the potential to decrease the risks of polluting N leaching in areas of intensive agriculture.

## 5 Conclusions

This study presents the responses of SOC and crop yield to straw incorporation under different farming management practices in various edaphic and climate regions in China. Compared with straw removal, straw incorporation significantly sequestered SOC at the rate of 0.35 Mg C ha$^{-1}$ yr$^{-1}$, increased crop yield by 13.4% and had a SOC conversion efficiency of 16% across the whole of China. A coupled benefit of straw incorporation at 3 Mg C ha$^{-1}$ yr$^{-1}$ with the mineral N rate of 200−400 kg N ha$^{-1}$ yr$^{-1}$ was exhibited




to be the best combination for farmers to use with crop yield increased by 32.7% and SOC sequestrated by the rate of 0.85 Mg C ha$^{-1}$ yr$^{-1}$. Straw incorporation achieved a higher SOC sequestration rate and crop yield increment in clay soils under high cropping intensities and in cold and humid area like Northeast China. As straw incorporation progressed, the SOC accrual rate declined and then stabilized;

5  crop yield responses increased and peaked at around 15-year and then declined. Our study confirmed that straw incorporation did create a positive feedback loop of SOC enhancement together with increased crop production, and the positive role of straw incorporation can play in China and global sustainable agriculture.

### 6 Data availability

10  Data are available by direct request to the corresponding author.

### 7 Author contributions

Fanqiao Meng, Xiao Han and Cong Xu conceived and designed the study. Xiao Han, Xiaojie Wang and Cong Xu carried the data collection and analysis. Xiao Han wrote the original draft. Cong Xu, Jennifer A. J. Dungait, Roland Bol, Fanqiao Meng and Wenliang Wu reviewed and revised the draft.

### 15  8 Competing interests

The authors declare that they have no conflict interest.

### 9 Disclaimer

Funders had no role in conceiving the study, collection and analysis of data or manuscript preparation.

### 10 Acknowledgements

20  This study was financially supported by the National Key Research and Development Program of China (Grant no.: 2016YFD0800100). This work was supported as part of Rothamsted Research's Institute Strategic Program–Soil to Nutrition (BB/PO1268X/1) funded by the UK Biotechnology and Biological Sciences Research Council. We thank the anonymous referees for their helpful comments and suggestions that greatly improved the manuscript.

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





**Table 1: Classification of categorical variables used as explanatory factors.**

| Categorical Variable | Level 1 | Level 2 | Level 3 | Level 4 |
|---|---|---|---|---|
| Soil texture | Clay | Loam | Silt loam | Sandy loam |
| Mean annual temperature (°C) | < 10 | 10−18 | > 18 | |
| Mean annual precipitation (mm) | < 600 | 600−1000 | > 1000 | |
| Experimental duration [a] (years) | 3−10 | 11−20 | > 20 | |
| Experimental duration [b] (years) | 3−5 | 6−10 | 11−15 | > 15 |
| Straw-C (Mg C ha$^{-1}$ yr$^{-1}$) | < 1.5 (Low) | 1.5−3 (Middle) | > 3 (High) | |
| N fertilizer (kg N ha$^{-1}$ yr$^{-1}$) | 0−200 (Low) | 200−400 (Middle) | 400−600 (High) | |

[a] experiment duration for the response of soil organic carbon to straw incorporation.
[b] experiment duration for the response of crop yield to straw incorporation.





**Table 2: Basic information on agricultural regions in the current analysis.**

| Region | Province | Crop frequency (season yr$^{-1}$) | Major crop | MAP (mm) | MAT (°C) |
|---|---|---|---|---|---|
| NEC | Heilongjiang, Jilin, Liaoning | Single | maize, soybean, wheat | 450−716 | 0−8.1 |
| NC | Beijing, Hebei, Henan, Shandong, Shanxi, Anhui (north region) | Double | maize, wheat | 455−821 | 7.3−14.8 |
| NWC | Shanxi, Gansu, Qinghai, Xinjiang | Single/Double | maize, wheat | 117−632 | 5.7−13 |
| SC | Jiangsu, Anhui (central and south region), Hubei, Hunan, Zhejiang, Shanghai, Guangxi, Chongqing, Sichuan, Jiangxi, Fujian | Double/Triple | maize, wheat, rice, rapeseed | 1038−1795 | 14.8−19.5 |

NEC: Northeast China; NC: North China; NWC: Northwest China; SC: South China.

MAP: Mean annual precipitation (mm); MAT: Mean annual temperature (°C).





**Table 3: Responses of soil organic carbon (SOC) (Mg C ha⁻¹ yr⁻¹) and crop yield (%) to straw incorporation over straw removal under different levels of straw C and N fertilizer input.**

| | N input rate (kg N ha⁻¹ yr⁻¹) | Straw-C input rate (Mg C ha⁻¹ yr⁻¹) | | | | | | | Impact | | | |
|---|---|---|---|---|---|---|---|---|---|---|---|---|
| | | nᵃ | < 1.5 | n | 1.5~3 | n | > 3 | Mean | n | S | N | S × N |
| Annual SOC sequestration rate (Mg C ha⁻¹ yr⁻¹) | 0–200 | 34 | 0.18 (0.14~0.23) | 17 | 0.27 (0.21~0.34) | 10 | 0.52 (0.42~0.61) | 0.27 (0.22~0.32) | 61 | * | ** | * |
| | 200–400 | 19 | 0.23 (0.16~0.32) | 15 | 0.42 (0.31~0.56) | 14 | 0.85 (0.54~1.15) | 0.46 (0.35~0.59) | 48 | | | |
| | >400 | 1 | 0.18 | 5 | 0.83 (0.72~0.94) | 4 | 0.64 (0.51~0.77) | 0.69 (0.53~0.81) | 10 | | | |
| | Mean | 54 | 0.20 (0.16~0.25) | 37 | 0.41 (0.33~0.49) | 28 | 0.70 (0.53~0.88) | | | | | |
| Year-around crop yield increment (%) | 0–200 | 20 | 14.0 (6.5~23.6) | 12 | 1.8 (-3.2~7.2) | 6 | 24.3 (13.0~37.8) | 11.5 (6.2~18.0) | 38 | * | | |
| | 200–400 | 9 | 12.1 (6.9~17.7) | 8 | 15.7 (6.1~34.6) | 6 | 32.7 (17.9~56.4) | 18.4 (11.9~27.6) | 23 | | | |
| | >400 | 1 | 71.2 | 3 | 5.2 (1.4~12.9) | 0 | – | 18.8 (1.6~54.2) | 4 | | | |
| | Mean | 30 | 15.0 (9.1~22.5) | 23 | 6.9 (2.3~14.5) | 12 | 28.4 (18.6~40.9) | | | | | |

ᵃ the number of observations.

\* and \*\* represent 0.05 and 0.01 significance levels, respectively.

S, N and S × N represent straw C input rate, N input rate and the interaction of the two, respectively.





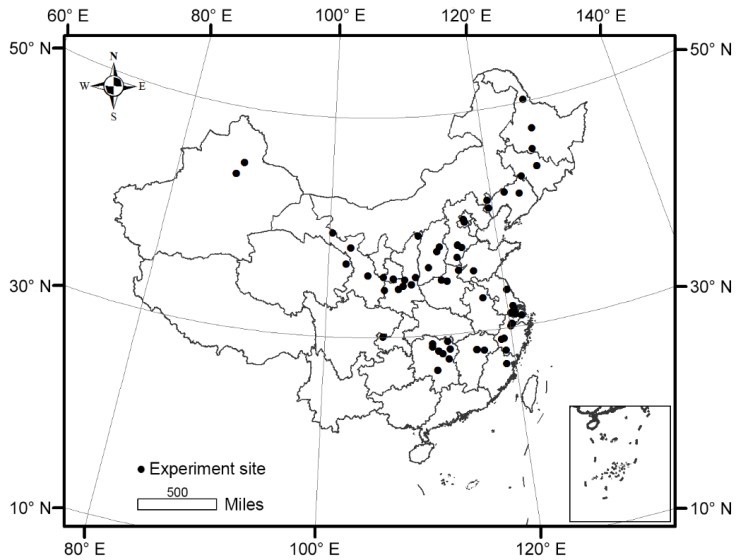

**Figure 1: Locations of the long-term experiment sites in China.**



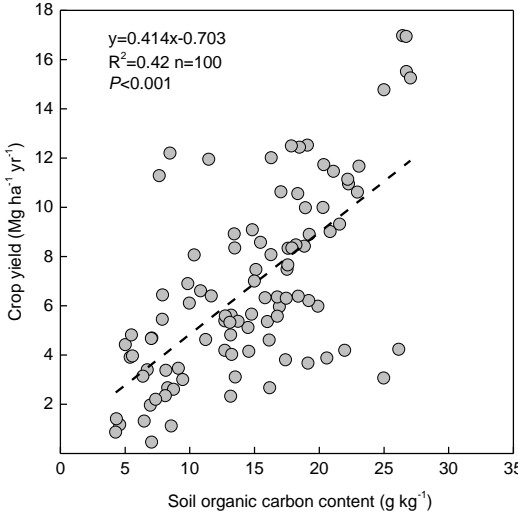

**Figure 2: Relationship between crop yield and soil organic carbon content.**




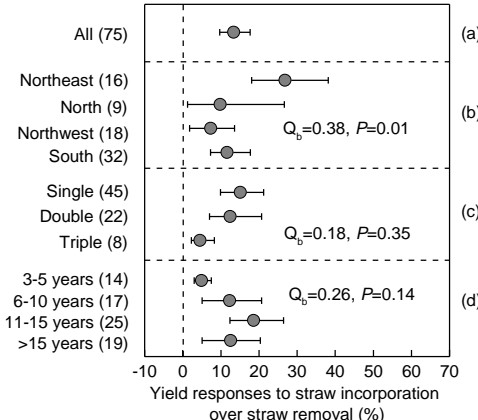

**Figure 3: Responses of crop yield to straw incorporation compared with straw removal (a), categorized into (b) region, (c) crop frequency and (d) experiment duration. Yield responses are expressed as the relative increase (%) compared with control (straw removal) with 95% confidence intervals represented by the error bars. Numbers of paired observations are in parentheses. Between-group heterogeneity ($Q_b$) and the probability ($P$) were used to describe statistical differences in yield responses between different levels of the categorized factors.**



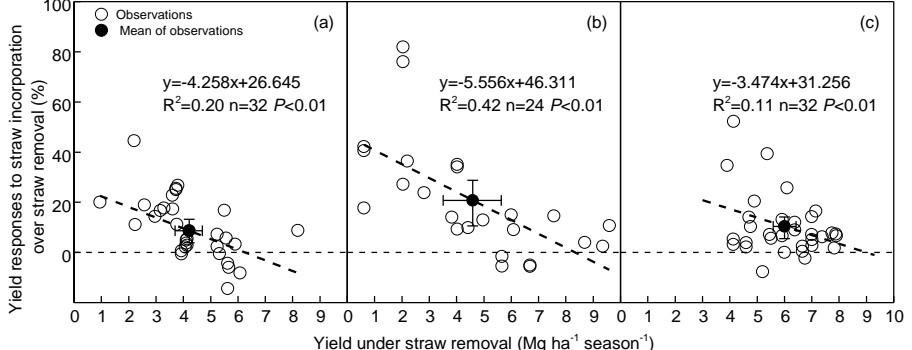

**Figure 4: Relationship between crop yield responses to straw incorporation and control yield under straw removal for the crop of (a) wheat, (b) maize and (c) rice. Yield responses are expressed as the relative increase (%) compared with control (straw removal). Error bars in horizontal and vertical directions represent 95% confidence intervals of the control yield and yield responses, respectively.**





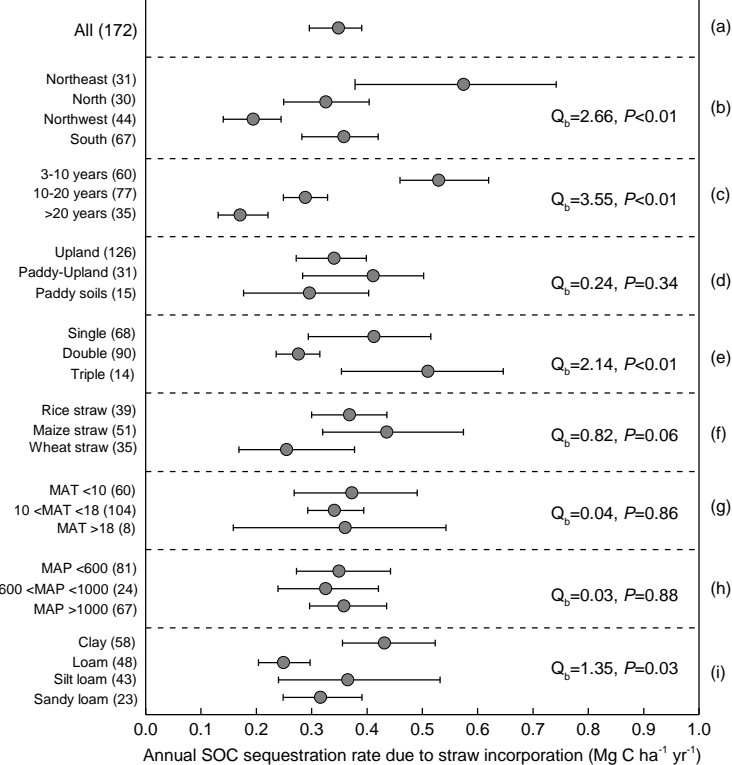

**Figure 5: Responses of soil organic carbon (SOC) to straw incorporation compared with straw removal (a), categorized into (b) region, (c) experiment duration, (d) land use, (e) crop frequency (season yr⁻¹), (f) straw type, (g) mean annual temperature (MAT), (h) mean annual precipitation (MAP) and (i) soil texture. SOC responses are expressed as the average annual SOC sequestration rate (Mg C ha⁻¹ yr⁻¹) with 95% confidence intervals represented by the error bars. Numbers of paired observations are in parentheses. Between-group heterogeneity ($Q_b$) and the probability ($P$) were used to describe statistical differences of SOC responses between different levels of the categorized factors.**



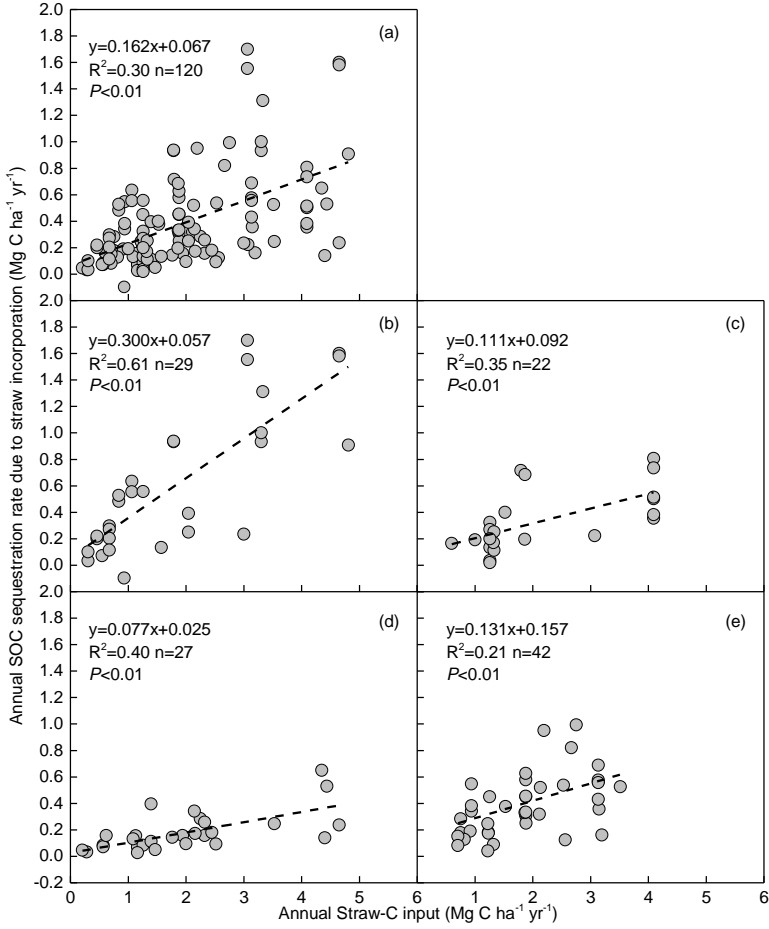

**Figure 6: Relationships between annual SOC sequestration rate and straw carbon input for (a) national scale, (b) Northeast China, (c) North China, (d) Northwest China and (e) South China.**





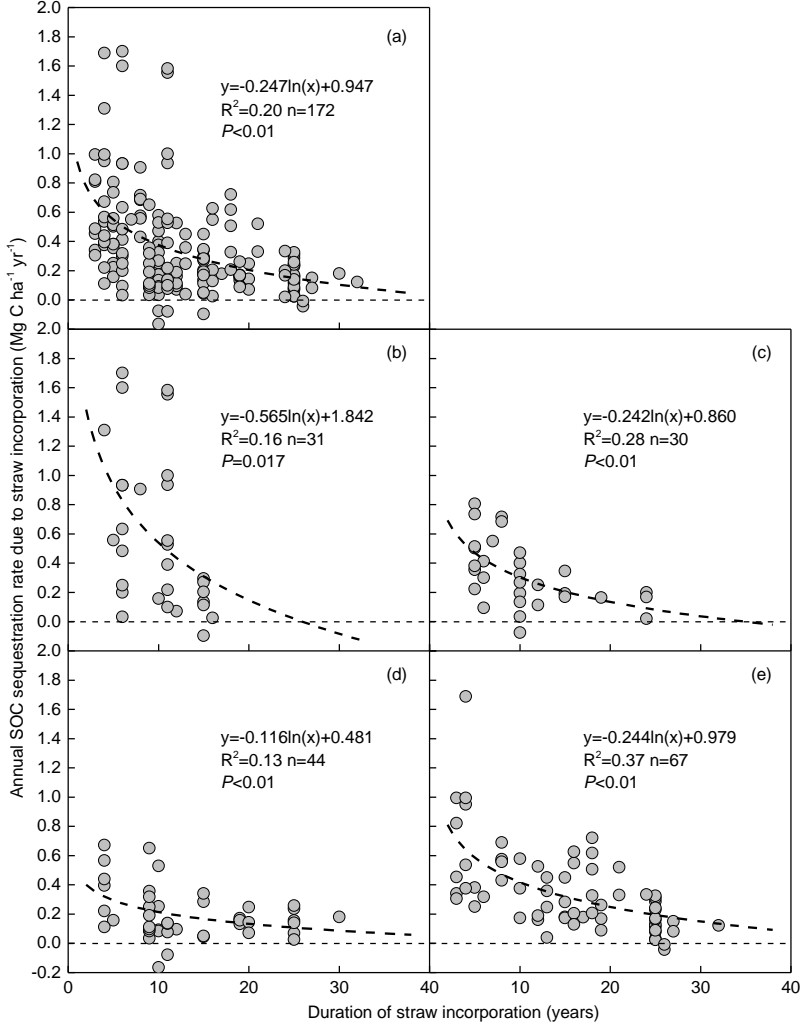

**Figure 7: Relationships between annual soil organic carbon sequestration rate and straw incorporation duration for (a) national scale, (b) Northeast China, (c) North China, (d) Northwest China and (e) South China.**