# Peer review of "Straw incorporation increases crop yield and soil organic carbon sequestration but varies under different natural conditions and farming practices in China: A system analysis"

_Biogeosciences, 2017_

## Referee Comment (RC1) · Anonymous Referee #1 · 25 Dec 2017

The authors present a meta-analysis about the effects of straw incorporation on crop production and SOC sequestration. The methods are technically sound. The authors also consider the effect of climate, straw carbon input, N fertilizer, and duration. This paper confirmed that straw incorporation did create a positive feedback loop of SOC enhancement together with increased crop production which is of great practical significance to agricultural management. However, I think that there are some part to be improved. Please edit closely for English. The sentences are often very long (even 5 lines) and, thus, difficult to follow and absorb immediately (i.e. P10 (line 21-26)). There

were many repeats of results in the discussion section. I hope that authors could improve it. 3.1 Why only consider the impact of N and K, how the effect of P fertilizer? In the result part, I suggest that authors deleted the range (i.e. range 2.3%–14.5%) P6 line 24-26 "with high levels of straw input corresponding to mean increases of 28.4% (range 18.6%......(mean 6.9%, range 2.3%–14.5%) straw input (Table 3)." 28.4% of what, I think it is of crop yield. P6 line 19-22"Meanwhile, yield increases greatly varied between crops: 8.7% (range 4.1%–20 13.5%)......the yield response to straw incorporation became smaller (Fig. 4)." I don't understand the means of this sentence. Here, the yield increase refer to the straw incorporation or control? And could you explain it in the discussion part? P6 line 26 Crop yield responses generally increase ......, delete the "responses" P8 line 9-11: This yield increase is similar in magnitude to a recent globa...... those of the EU (6% increase; Lehtinen et al., 2014)." Could you explain the reason for this differences? P8 line 19 and the greater the annual straw-C input.... Change "and" to "And" P8 line 26-29: "Furthermore, N fertilizer addition can enhance both above and belowground biomass production (Ladha et al., 2011; Neff et al., 2002; Kuzyakov and Domanski, 2000), increasing the input of crop roots to stable SOC pools (Gong et al., 2012)." I think this sentence should be improved. P8 line 34 "...straw incorporation effect on SOC was observed between the four..." change "between" to "among" P 9 line1-2 "compared to large-scale increases in SOC in the majority of croplands in NC, NWC and SC." Here, the SOC means SOC stocks? P 9 line3: According to a farmer survey across China carried out by (Zhang et al., 2017), Change it to "According to a farmer survey across China carried out by Zhang et al. (2017) P 9 line6: "The impact of land use, MAT, and MAP on straw-induced SOC sequestration was not ........ and Huang et al. (2012)." to "The impacts of land use, MAT, and MAP on straw-induced SOC sequestration were not statistically significant (Fig. 5; P > 0.05), in agreement with the previous meta-analyses of Liu et al. (2014) and Huang et al. (2012)." P 9 line9: "this wetting and drying cycles" change "this" to "these" P 9 line8-11: "Since alternative wetting and drying has been widely....leads to a less stable form of SOC in paddy soils (Cui et al., 2012)." This is a long sentence, and change

"increases", "leads" to "increase" and "lead". P 9 line 23-27: "The lower estimates reported in previous studies focused on shorter time periods........were included in the analysis by Wang et al. (2015) and Huang et al. (2013)." This is a long sentence. P9 line 28: "result of " to "result in" P9 line 35-36: Change "Straw incorporation does also reduce" to "Straw incorporation also reduces" P10 (line 7-11) "Our analysis did observe a stro........(Fig. 4). This observation agre..........those of wheat or barley." I think that these two sentences are repeated with each other. P10 (line 10-12): Generally, the maize yields is the highest among the three types of crop (wheat, rice and maize). This is not only just because the temperature and precipitation. What's more, the result in the paper found that climate has no significant effect on the response of SOC to straw incorporation. P10 (line 22) "In China, the areas where triple cropping was adopted usually received adequate rainfall (MAP > 1000 mm, Table S1)" I think the yield increase is due to the temperature and precipitation. P11 (line 5) "crop yield responses increased and peaked at around 15-year and then declined." Delete "responses" P11 (line 7-8) Change "and the positive role of straw incorporation can play in China and global sustainable agriculture." to "and the positive role of straw incorporation playing in China and global sustainable agriculture." Table 2 Add the information about the soil type of the different regions. "Table 1:" to "Table 1."

---

## Referee Comment (RC2) · Anonymous Referee #2 · 8 Jan 2018

The authors conducted a meta-analysis to examine the impact of straw incorporation on SOC sequestration in China. Their analysis identified the best combination of straw incorporation strategy and quantified the impact of different approaches. They also provided a general timeline for the response of SOC and crop productivity respectively.

The collected dataset has many missing values and the authors have made multiple assumptions to fill in the blanks, including using empirical functions and coefficients, it would be better if the authors can provide some kind of uncertainty analysis to ensure their results still holds under these noisy extrapolations.

[Figure]

The study focused on SOC changes in the top 20cm soil, does this include organic horizon or is mineral soil only? Please make the distinction. Also it would be curious to see data for SOC below this depth.

The language can be improved as well, and the text can be shortened and be more succinct if collapsing some of the results and discussions that are repetitive. Overall, the study explores an interesting topic and provided quantitative proof of the impact of straw incorporation on SOC sequestration of soils. The analysis approach is appropriate. Some implications of these findings are lacking in current version, it would be good to expand on.

---

## Author Comment (AC1) · 27 Jan 2018

Dear respected Referee #1 (R#1), Thank you so much for your valuable comments and helpful suggestions. We have fully studied your review and revised the MS substantially. In your several comments, it seemed that "crop yield response" has been confused with "crop yield". In our study, "crop yield response" is the change of crop yield due to farming practice, such as straw incorporation.

R#1: The authors present a meta-analysis about the effects of straw incorporation

on crop production and SOC sequestration. The methods are technically sound. The authors also consider the effect of climate, straw carbon input, N fertilizer, and duration. This paper confirmed that straw incorporation did create a positive feedback loop of SOC enhancement together with increased crop production which is of great practical significance to agricultural management. However, I think that there is some part to be improved. Please edit closely for English. The sentences are often very long (even 5 lines) and, thus, difficult to follow and absorb immediately (i.e. P10 (line 21-26)). There were many repeats of results in the discussion section. I hope that authors could improve it.

[Responses]: Thank you for the encouragement. We'll improve the Discussion section to avoid the repetitive sentences of results. The language will be further refined by our author, Professor Jennifer Dungait from Rothamsted and Professor Roland Bol from Juelich Center.

R#1: 3.1 Why only consider the impact of N and K, how the effect of P fertilizer?

[Responses]: Actually, we also considered the effect of P fertilizer in the study, which was stated in section 2.4. However, after the stepwise regression analysis has been finished, only the variables of SOC, N, and K were kept and P was excluded. We will clarify this in the MS.

R#1: In the result part, I suggest that authors deleted the range (i.e. range 2.3%– 14.5%)

[Responses]: Agreed and will revise accordingly.

R#1: P6 line 24-26 "with high levels of straw input corresponding to mean increases of 28.4% (range 18.6%......(mean 6.9%, range 2.3%–14.5%) straw input (Table 3)." 28.4% of what, I think it is of crop yield.

[Responses]: Here, the "28.4%" refers to the crop yield response at the high level of straw input. We'll improve these sentences to avoid misunderstanding.

R#1: P6 line 19-22 "Meanwhile, yield increases greatly varied between crops: 8.7% (range 4.1%–20 13.5%)......the yield response to straw incorporation became smaller (Fig. 4)." I don't understand the means of this sentence. Here, the yield increase refers to the straw incorporation or control? And could you explain it in the discussion part?

[Responses]: The yield increase/response referred to the yield increase under straw incorporation relative to the control (straw removal). The sentences will be revised as "Yield response to straw incorporation was greater when the yield of control (straw removal, or background crop yield) was low and, as the yield of control increased, the yield response became smaller (Fig. 4)". The discussion will be added as suggested.

R#1: P6 line 26 Crop yield responses generally increase......, delete the "responses".

[Responses]: As responded above, "responses" should be used instead of deleted, to reflect the interactive effect of straw×mineral N fertilization.

R#1: P8 line 9-11: This yield increase is similar in magnitude to a recent global...... those of the EU (6% increase; Lehtinen et al., 2014)." Could you explain the reason for this differences?

[Responses]: The reason might be the different climate zones in the EU and our study. Specifically, experimental sites in the Mediterranean, a typical climate zone of Europe, accounted for 25% of the database in the study of Lehtinen et al. (2014). These sites mostly exhibited a yield decrease under straw incorporation, thus lowered the mean yield responses of the EU. We'll address in the revised Discussion.

Lehtinen, T., Schlatter, N., Baumgarten, A., Bechini, L., Kruger, J., Grignani, C., Zavattaro, L., Costamagna, C., and Spiegel, H.: Effect of crop residue incorporation on soil organic carbon and greenhouse gas emissions in European agricultural soils, Soil Use Manage, 30, 524−538, doi:10.1111/sum.12151, 2014.

R#1: P8 line 19 and the greater the annual straw-C input... Change "and" to "And"

[Responses]: Agreed and will revise accordingly.

R#1: P8 line 26-29: "Furthermore, N fertilizer addition can enhance both above and belowground biomass production (Ladha et al., 2011; Neff et al., 2002; Kuzyakov and Domanski, 2000), increasing the input of crop roots to stable SOC pools (Gong et al., 2012)." I think this sentence should be improved.

[Responses]: Agreed. This sentence will be revised as "Furthermore, N fertilizer addition can enhance both above and belowground biomass production (Ladha et al., 2011; Neff et al., 2002; Kuzyakov and Domanski, 2000), which will result in a higher crop yield and the improvement of SOC (Gong et al., 2012)".

R#1: P8 line 34 "…straw incorporation effect on SOC was observed between the four…" change "between" to "among"

[Responses]: Agreed and will revise accordingly.

R#1: P 9 line1-2 "compared to large-scale increases in SOC in the majority of croplands in NC, NWC, and SC." Here, the SOC means SOC stocks?

[Response]: Yes, the SOC here means SOC stocks. We'll add this to the revised MS.

R#1: P 9 line3: According to a farmer survey across China carried out by (Zhang et al.,2017), Change it to "According to a farmer survey across China carried out by Zhang et al. (2017)

[Responses]: Thanks, and will revise accordingly.

R#1: P 9 line6: "The impact of land use, MAT, and MAP on straw-induced SOC sequestration was not……and Huang et al. (2012)." to "The impacts of land use, MAT, and MAP on straw-induced SOC sequestration were not statistically significant (Fig. 5; $P > 0.05$), in agreement with the previous meta-analyses of Liu et al. (2014) and Huang et al. (2012)."

[Responses]: Agreed and will revise accordingly.

R#1: P 9 line9: "this wetting and drying cycles" change "this" to "these"

[Figure]

[Responses]: Agreed and will revise accordingly.

R#1: P 9 line8-11: "Since alternative wetting and drying has been wide.......leads to a less stable form of SOC in paddy soils (Cui et al., 2012)." This is a long sentence, and change "increases", "leads" to "increase" and "lead".

[Responses]: Agreed and we are going to refine all these long sentences.

R#1: P 9 line 23-27: "The lower estimates reported in previous studies focused on shorter time periods......were included in the analysis by Wang et al. (2015) and Huang et al. (2013)." This is a long sentence.

[Responses]: Same as above.

R#1: P9 line 28: "result of" to "result in"

[Responses]: Agreed and will revise accordingly.

R#1: P9 line 35-36: Change "Straw incorporation does also reduce" to "Straw incorporation also reduces"

[Responses]: Agreed and will revise accordingly.

R#1: P10 (line 7-11) "Our analysis did observe a stro......(Fig. 4). This observation agrees.......those of wheat or barley." I think that these two sentences are repeated with each other.

[Responses]: Agreed and these two sentences will be revised as "The beneficial effect on yield response was more for maize (20.8%) than that for wheat (8.7%) (Fig. 4), which agrees with the study of Hijbeek et al. (2017).".

R#1: P10 (line 10-12): Generally, the maize yields is the highest among the three types of the crop (wheat, rice, and maize). This is not only just because of the temperature and precipitation. What's more, the result in the paper found that climate has no significant effect on the response of SOC to straw incorporation.

[Responses]: As mentioned above, here we discussed the relative increase of maize yield induced by straw incorporation, instead of the absolute maize yield. We found that under straw incorporation, yield increase was higher for maize than for wheat. This is most likely due to hotter and humid condition in the maize season than that in the wheat season (Tan et al., 2017). Hotter and humid condition stimulated the straw decomposition and released fast and more nutrients to crop production (Hartmann et al., 2014; Ladha et al., 2011). Here, we focused on crop yield responses, rather than the SOC responses.

Hartmann, T. E., Yue, S., Schulz, R., Chen, X., Zhang, F., and Müller, T.: Nitrogen dynamics, apparent mineralization and balance calculations in a maize–wheat double cropping system of the North China Plain, Field Crop Res, 160, 22–30, doi:10.1016/j.fcr.2014.02.014, 2014.

Ladha, J. K., Reddy, C. K., Padre, A. T., and van Kessel, C.: Role of nitrogen fertilization in sustaining organic matter in cultivated soils, J Environ Qual, 40, 1756−1766, doi:10.2134/jeq2011.0064, 2011.

Tan, Y. C., Xu, C., Liu, D. X., Wu, W. L., Lal, R., and Meng, F. Q.: Effects of optimized N fertilization on greenhouse gas emission and crop production in the North China Plain, Field Crop Res, 205, 135−146, doi:10.1016/jScr.2017.01.003, 2017.

R#1: P10 (line 22) "In China, the areas where triple cropping was adopted usually received adequate rainfall (MAP > 1000 mm, Table S1)" I think the yield increase is due to the temperature and precipitation.

[Responses]: We agreed that straw incorporation could improve crop yield by improving soil water retention as well as reducing abrupt fluctuations in soil temperature. The low soil water availability and low temperature are likely to be limiting factors for crop yield especially in the single cropping areas (MAP: 117∼716 mm, MAT: 0.9∼11.5 degrees centigrade; Table S1) compared with that in triple cropping areas (MAP > 1000 mm, MAT: 14.8∼17.6 degrees centigrade; Table S1). Thus, straw incorporation might

contribute more benefits for crop production in the single cropping areas compared to that in triple cropping areas. We'll revise the manuscript.

R#1: P11 (line 5) "crop yield responses increased and peaked at around 15-year and then declined." Delete "responses"

[Responses]: As responded above, the crop yield response was the effect size to explore the effect of straw incorporation on crop yield. Here, it was the crop yield increment increased and peaked at around 15-year, so "responses" should be kept.

R#1: P11 (line 7-8) Change "and the positive role of straw incorporation can play in China and global sustainable agriculture." to "and the positive role of straw incorporation playing in China and global sustainable agriculture."

[Responses]: Agreed and will revise accordingly.

R#1: Table 2 Add the information about the soil type of the different regions.

[Responses]: Agreed and will add soil types in the revised Table.

R#1: "Table 1:" to "Table 1."

[Responses]: Agreed and will revise accordingly.

---

## Author Comment (AC2) · 27 Jan 2018

**Dear respected Referee #2 (R#2),**

**Your guiding comments and suggestions are highly appreciated. Our responses are listed below.**

R#2: The authors conducted a meta-analysis to examine the impact of straw incorporation on SOC sequestration in China. Their analysis identified the best combination of straw incorporation strategy and quantified the impact of different approaches. They also provided a general timeline for the response of SOC and crop productivity respectively.

The collected dataset has many missing values and the authors have made multiple assumptions to fill in the blanks, including using empirical functions and coefficients, it would be better if the authors can provide some kind of uncertainty analysis to ensure their results still holds under these noisy extrapolations.

**[Responses]: We agree with the reviewer that these assumptions indeed decrease the robustness and certainty of our study. Actually, during the MS preparation, we conducted an uncertainty analysis to test whether these estimations by using different empirical functions and coefficients would affect our results. We listed our results in the table below, which reported the comparison results of SOC responses with different BD (Table 1) and straw-C (Table 2) estimation approaches. The overall SOC response was 0.35 (95% CI, 0.31~0.40) Mg C ha$^{-1}$ yr$^{-1}$ in the Current scenario of BD estimation, while the SOC response was 0.33 (0.29~0.42), 0.35 (0.29~0.40), 0.35 (0.31~0.40) Mg C ha$^{-1}$ yr$^{-1}$ in scenario A, B and C, respectively. The relationship between SOC response and straw-C input in Current scenario was y=0.162x+0.067 (R$^2$=0.30 n=120 $P$<0.01), while the relationship was y=0.170x+0.059 (R$^2$=0.30 n=120 $P$<0.01) in scenario I of straw-C estimation. We did not report the findings in the 1$^{st}$ version of MS because the estimation approaches gave very similar results without significant differences (P >0.05). However, as the reviewer suggested, in the revised manuscript we will add the uncertainty analysis in the M&M and Discussion Sections.**

**Table 1. Comparison of the SOC responses using different BD estimation approaches.**

| Scenario | BD estimation approach | Annual SOC sequestration rate (Mg C ha$^{-1}$ yr$^{-1}$) | | | |
|---|---|---|---|---|---|
| | | All | Upland | Paddy | Paddy-Upland |
| Current | Eq. (1) for paddy or paddy–upland soil BD, Eq (2) for upland; | 0.35 (0.31~0.40) | 0.34 (0.28~0.41) | 0.30 (0.19~0.42) | 0.41 (0.33~0.51) |
| A | All the missed BD was estimated by Eq (1) | 0.33 (0.29~0.42) | 0.32 (0.26~0.38) | 0.30 (0.18~0.42) | 0.37 (0.26~0.51) |
| B | All the missed BD was estimated by Eq (2) | 0.35 (0.29~0.40) | 0.33 (0.26~0.40) | 0.32 (0.21~0.50) | 0.41 (0.27~0.54) |
| C | All the missed BD was estimated by Eq (3) | 0.35 (0.31~0.40) | 0.35 (0.28~0.42) | 0.29 (0.18~0.40) | 0.39 (0.31~0.49) |

Note: the estimation equations for BD:

Eq. (1): $BD = -0.22 \times \ln(SOC) + 1.78$ $\qquad$ (Pan et al., 2003)

Eq. (2): $BD = 1.377 \times \text{Exp}(-0.0048 \times SOC)$ $\qquad$ (Song et al., 2005)

Eq. (3): $BD = -0.247 \times \ln(SOC) + 1.867$

Eq. (3) was derived from the empirical relationship between SOC content and BD based on 239 analytical samples in our database.

**Table 2. Comparison of the SOC responses to straw-C input from different straw-C estimation approaches.**

| Scenario | Carbon concentration (%) of crop straw | Relationship between SOC responses (y; Mg C ha$^{-1}$ yr$^{-1}$) and straw carbon input (x; Mg C ha$^{-1}$ yr$^{-1}$) for national scale |
|---|---|---|
| Current | Wheat: 39.9%; Maize: 44.4%; Rice: 41.8%; (NATEC, 1999) | y=0.162x+0.067, R$^2$=0.30 n=120 $P$<0.01 |
| I | 40% for all the straw type; (Liu et al., 2014) | y=0.170x+0.059, R$^2$=0.30 n=120 $P$<0.01 |

Liu, C., Lu, M., Cui, J., Li, B., and Fang, C.: Effects of straw carbon input on carbon dynamics in agricultural soils: a meta-analysis, Glob Change Biol, 20, 1366-1381, doi:10.1111/gcb.12517, 2014.

National Agro-Tech Extension Center (NATEC): Chinese Organic Fertilizer Handbook, Chinese Agricultural Press, Beijing, China, 1999 (in Chinese).

Pan, G., Li, L., Wu, L., and Zhang, X.: Storage and sequestration potential of topsoil organic carbon in China's paddy soils, Glob Change Biol, 10, 79–92, doi:10.1111/j.1365-2486.2003.00717.x, 2003.

Song, G. H., Li, L. Q., Pan, G. X., and Zhang, Q.: Topsoil organic carbon storage of China and its loss by cultivation, Biogeochemistry, 74, 47−62, doi:10.1007/s10533-004-2222-3, 2005.

R#2: The study focused on SOC changes in the top 20 cm soil, does this include

organic horizon or is mineral soil only? Please make the distinction. Also it would be curious to see data for SOC below this depth.

**[Responses]: In our database, the SOC content of the top 20 cm ranged from 0.16% to 3.21%, i.e., mostly mineral soils. Even in northeast China, the studies adopted in our study are on agricultural soils which have been reclaimed more than 5 years, and with SOM low than 5%. We'll clarify this in the revised MS. Previous experiments conducted in China mostly are on 0-20 cm, although there several good studies revealing the significant value of deeper SOC. We will also address this in the revised MS.**

R#2: The language can be improved as well, and the text can be shortened and be more succinct if collapsing some of the results and discussions that are repetitive.

**[Responses]: Thank you for this good advice. Our revised MS will also be further refined by our authors of Jennifer Dungait and Roland Bol, considering of your advice.**

R#2: Overall, the study explores an interesting topic and provided quantitative proof of the impact of straw incorporation on SOC sequestration of soils. The analysis approach is appropriate. Some implications of these findings are lacking in the current version, it would be good to expand on.

**[Responses]: Agreed. The important implications will be added to the Discussion and Conclusion Sections.**